# Comparison of Structural Features of CRISPR-Cas Systems in Thermophilic Bacteria

**DOI:** 10.3390/microorganisms11092275

**Published:** 2023-09-10

**Authors:** Chuan Wang, Yuze Yang, Shaoqing Tang, Yuanzi Liu, Yaqin Wei, Xuerui Wan, Yajuan Liu, Zhao Zhang, Yongjie Sunkang

**Affiliations:** 1College of Veterinary Medicine, Gansu Agricultural University, Lanzhou 730070, China; yuanzi.liu@meili-oh.cn (Y.L.); wanxr@gsau.edu.cn (X.W.); 18893114407@163.com (Y.L.); zzhao0829@163.com (Z.Z.); 2Beijing Animal Husbandry Station, Beijing 100070, China; yyz84929056@126.com (Y.Y.); tangshaoqingbj@163.com (S.T.); 3Key Laboratory of Microbial Resources Exploitation and Application of Gansu Province, Center for Anaerobic Microbes, Institute of Biology, Gansu Academy of Sciences, Lanzhou 730000, China; weiyq11@lzu.edu.cn

**Keywords:** CRISPR-Cas systems, CRISPR repeats, optimum temperature, spacer, thermophilic bacteria

## Abstract

The clustered regularly interspaced short palindromic repeat (CRISPR) is an adaptive immune system that defends most archaea and many bacteria from foreign DNA, such as phages, viruses, and plasmids. The link between the CRISPR-Cas system and the optimum growth temperature of thermophilic bacteria remains unclear. To investigate the relationship between the structural characteristics, diversity, and distribution properties of the CRISPR-Cas system and the optimum growth temperature in thermophilic bacteria, genomes of 61 species of thermophilic bacteria with complete genome sequences were downloaded from GenBank in this study. We used CRISPRFinder to extensively study CRISPR structures and CRISPR-*associated* genes (*cas*) from thermophilic bacteria. We statistically analyzed the association between the CRISPR-Cas system and the optimum growth temperature of thermophilic bacteria. The results revealed that 59 strains of 61 thermophilic bacteria had at least one CRISPR locus, accounting for 96.72% of the total. Additionally, a total of 362 CRISPR loci, 209 entirely distinct repetitive sequences, 131 *cas* genes, and 7744 spacer sequences were discovered. The average number of CRISPR loci and the average minimum free energy (MFE) of the RNA secondary structure of repeat sequences were positively correlated with temperature whereas the average length of CRISPR loci and the average number of spacers were negatively correlated. The temperature did not affect the average number of CRISPR loci, the average length of repeats, or the guanine-cytosine (GC) content of repeats. The average number of CRISPR loci, the average length of the repeats, and the GC content of the repeats did not reflect temperature dependence. This study may provide a new basis for the study of the thermophilic bacterial adaptation mechanisms of thermophilic bacteria.

## 1. Introduction

The clustered regularly interspaced short palindromic repeats (CRISPR) structure was first discovered in the nucleotide sequence of the *Escherichia coli iap* gene [1]. This structure was subsequently discovered in more bacteria and archaea, and in 2002 it was formally named CRISPR [2]. The CRISPR of *Streptococcus thermophilus* is involved in defending against exogenous genetic factors (phages) [3]. The structure of the CRISPR locus consists of an average of approximately 32 base pairs (bp) of repeat sequences, spacer sequences, conserved lead sequences of hundreds of bp, and CRISPR-associated (*cas*) genes [4]. The RNA of repeat sequences is repeated to create a stable stem-loop secondary structure which demonstrates compensatory mutations indicating structural stability. CRISPR repeat sequences are not structurally homogeneous and can be divided into distinct types based on sequence similarity and the ability to form stable secondary structures [5]. Spacer sequences are usually derived from mobile gene progenitors such as viruses, phages, and plasmids [6]. A conserved gene sequence called protospacer adjacent motif, significant in the CRISPR defense process, is adjacent to the homologous sequence protospacer upstream or downstream of the intervening sequence [7]. The lead sequence, which is relatively conserved within the same species, is an approximately 300–500 bp long adenine-thymine (AT)-rich sequence located at the 5′ end of the CRISPR locus and linked to a repeat sequence [2]. It has internal sequences that control CRISPR transcription and specifically recognize cas proteins [8]. CRISPR structures are accompanied by cas genes nearby. There are several types of *cas* genes, including *cas1*-*cas10* [9]. Almost every CRISPR structure has the *cas1* gene, which is regarded as the core gene of the CRISPR system [10]. The CRISPR-Cas system has two classes, six types, and thirty-three subtypes. Class 1 CRISPR–Cas systems have effector modules composed of multiple cas proteins, including types I, III, and IV. Class 2 CRISPR–Cas systems have a single, multidomain crRNA-binding protein that is functionally analogous to the entire effector complex of class 1, including types II, V, and VI [11]. The prokaryotic immunization process mediated by the CRISPR-Cas system can be divided into three phases—the CRISPR adaptation, the CRISPR expression, and the CRISPR interference phases [12]. Currently, the CRISPR-Cas system is mainly used for strain typing, gene editing, and other technologies [13,14]. Thermophilic bacteria are heat-resistant acidophilic bacteria and archaea that oxidize ferrous iron, elemental sulfur, reduced-state inorganic sulfides, and sulfide minerals in their environment and grow at or above 45 °C and at a pH below 3.0. These bacteria are commonly classified into moderate thermophiles (45–60 °C), thermophiles (60–80 °C), and hyperthermophiles (>80 °C) [15]. According to the CRISPR database (http://crispr.i2bc.paris-saclay.fr, accessed on 15 September 2022), 202 (87%) of 232 archaea with whole genome sequences and 3059 (45%) of 6782 bacteria were identified to have CRISPRs structures [16]. The prevalence of CRISPR-Cas systems in thermophiles is significantly higher than in mesophiles, according to the analysis of hundreds of different prokaryotic genomes [17]. The function of CRISPR may be related to the adaptation of organisms to high temperatures [2]. However, the correlation between CRISPR and the temperature of thermophilic bacteria is unclear. Therefore, in this study, a comparative analysis of the CRISPR structures of 61 thermophilic bacteria possessing whole genome sequences was conducted to investigate the structural differences of CRISPR loci in different thermophilic bacterial species and to examine the relationship between the CRISPR-Cas system and the optimum growth temperature in thermophilic bacteria.

## 2. Materials and Methods

### 2.1. Data Collection

Thermophilic bacteria were classified as moderate thermophiles, thermophiles, and hyperthermophiles according to their optimum growth temperature. The optimum growth temperature of moderate thermophiles, thermophiles, and hyperthermophiles was 45 to 60 ℃, 60 to 80 ℃, and above 80 °C, respectively. All genome sequences of thermophilic bacteria strains, including 16 moderate thermophiles, 40 thermophiles, and 5 hyperthermophiles, were retrieved and downloaded from the National Center for Biotechnology Information (NCBI) database (https://www.ncbi.nlm.nih.gov/, accessed on 1 June 2022). The specific information for the strains is shown in Appendix A.

### 2.2. CRISPR/Cas System Identification

Thermophilic bacteria CRISPR loci were searched by the CRISPRFinder server (https://crispr.i2bc.paris-saclay.fr/Server/, accessed on 5 September 2022) with default parameters (Last updated on 9 May 2017) [17]. The sequences in the range of 10,000 bp upstream and 10,000 downstream of flanking regions in base pairs (bp) for each analyzed CRISPR array were uploaded to the CRISPRCas Finder web server (https://crisprcas.i2bc.paris-saclay.fr/CrisprCasFinder/Index, accessed on 15 September 2022) to determine the presence and content of *cas* genes [18].

### 2.3. Bioinformatics Analysis of CRISPR/Cas System

Multiple sequence alignments (MSA) of all consensus direct repeats (CDRs), Cas1, and Cas2 were carried out by MEGA X software. The sequences of Cas1 and Cas2 are provided in the Appendix A. The multiple sequence alignments of protein sequences of Cas1 and Cas2 were performed using the MUSCLE algorithm, and then phylogenetic trees were constructed with the neighbor joining method by MEGA X and decorated by iTOL (https://itol.embl.de/, accessed on 31 August 2023). The G + C content of CDRs were analyzed with DNAstar software, and then the RNA secondary structures and minimum free energy (MFE) of each direct repeat sequence were predicted by the RNA fold web server (http://rna.tbi.univie.ac.at/cgi-bin/RNAWebSuite/RNAfold.cgi, accessed on 22 September 2022). The CDR sequences were further investigated with the CRISPRmap v2.1.3-2014 tool (http://rna.informatik.uni-freiburg.de/CRISPRmap, accessed on 18 September 2021), and CDRs conservation was analyzed by online Weblogo software (http://weblogo.berkeley.edu/, accessed on 2 October 2021). The statistical analysis of CRISPR spacers was performed by a script written in R language (https://cran.r-project.org/, accessed on 23 September 2022). The details of the code used for data analysis are provided in the Appendix A. 

## 3. Results

### 3.1. Genomic Distribution of CRISPR Structural Loci in Thermophilic Bacteria

All the genome sequences are available at the National Center for Biotechnology Information. As presented in Appendix A, 282 confirmed CRISPR loci and 80 questionable CRISPR loci were identified in the CRISPR loci analysis of 59 strains of thermophilic bacteria. Of the 61 thermophilic strains, 59 contained CRISPR loci (including questionable CRISPR loci), accounting for 96.72%; 57 thermophilic strains contained confirmed CRISPR loci, accounting for 93.44%. The average number of CRISPR loci for the 59 thermophilic strains containing CRISPR loci (including questionable CRISPR loci) and for the 57 containing identified CRISPR loci was 6.3 and 5.0, respectively. Two thermophiles, *Thermus thermophilus* JL-18 and *Thermus parvatiensis* strain RL, did not contain CRISPR structures in their genomic sequences. Among the 59 strains containing CRISPR loci (including questionable CRISPR loci), *Thermomonospora curvata* DSM 43183 had the most CRISPR structures with 20 CRISPR loci. The least number of CRISPR loci was observed in *Thermosynechococcus elongatus* BP-1, *Thermosynechococcus* sp. NK55, *Thermoanaerobacterium thermosaccharolyticum* M0795, *Thermosulfidibacter takaii* ABI70S6, and *Thermomicrobium roseum* DSM 5159, each with only one CRISPR locus. In 16 moderate thermophiles, 111 CRISPR loci were observed, with an average of 6.9 CRISPR loci per moderate thermophile. There were 206 CRISPR loci in 40 thermophiles, with an average of 5.15 CRISPR loci per thermophile. In total, 45 CRISPR loci were observed in 5 hyperthermophiles, with an average of 9 CRISPR loci per hyperthermophile. The length of CRISPR loci ranged from 74 to 14,581 bp, with an average length of 1446 bp per thermophilic bacteria. The longest CRISPR loci was observed in *Thermoanaerobacter* sp. X514 and the shortest CRISPR loci was *Thermobacillus composti* KWC4. The average length of CRISPR locus for moderate thermophiles, thermophiles, and hyperthermophiles was 1550.73 bp, 1548.77 bp, and 716.87 bp, respectively.

### 3.2. Distribution and Type of Cas Genes

*Cas* gene types of 55 thermophilic bacteria are depicted in Figure 1. The complete CRISPR-Cas system consisting of CRISPR structure and *cas* gene was identified in 55 thermophilic bacteria strains, accounting for 90.16%. The classification of the CRISPR-Cas system of these strains revealed that they belonged to types I, III, IV, and V; however, types II and VI were not present. The types of CRISPR-Cas system and the proportions of moderate thermophiles, thermophiles, and hyperthermophiles are exhibited in Table 1. The largest proportion of CRISPR-Cas system of thermophilic bacteria was type III, followed by type I, type IV, and type V, accounting for 49.62%, 45.03%, 4.58%, and 0.76%, respectively. There were five subtypes of type I: IA, IB, IC, ID, and IE, and four subtypes of type III: IIIA, IIIB, IIIC, and IIID. A total of 131 subtypes of CRISPR-Cas system were detected in 55 thermophilic bacteria strains, including 59 subtypes of type I (5 subtype IA, 45 subtype IB, 1 subtype IC, 2 subtype ID, and 6 subtype IE), 65 subtypes of type III (17 subtype IIIA, 21 subtype IIIB, 11 subtype III C, and 16 subtype IIID), 1 subtypes of type IV, and 6 subtypes of type V. Of these, the largest proportion of subtypes of CRISPR-Cas system was the subtype IB of thermophiles, accounting for 23.66%. The phylogenetic tree based on protein sequences of Cas1 and Cas2 implemented in the MEGA X and decorated by iTOL were presented in Figure 2. The phylogenetic tree established by Cas1 and Cas2 revealed that these proteins were highly conserved. Different genera of thermophiles were mixed, making it impossible to separate them or classify them according to their optimum growth temperature.

### 3.3. Repeat Sequence

The repeat sequences of 59 thermophilic bacteria strains are presented in Appendix A. In total, 362 repeat sequences from 59 thermophilic bacteria strains were identified. After eliminating the same sequence, we compared the obtained 209 different repeat sequences with the CRISPRmap database. The length and number of repeat sequences of different thermophiles are displayed in Table 2. The repeat sequences ranged in length from 23–37 bp with an average length of 29.5 bp, and the number of repeats varied from 2 to 220 with an average of 22.39. The average guanine-cytosine (GC) content of repeat sequences was 45.21%, with the highest being 92% and the lowest being 12%. Among 209 completely different repeat sequences, 187 formed the secondary structure. The highest MFE, the lowest MFE, and the average MFE of the RNA secondary structure was −0.20 kcal/mol, −15.50 kcal/mol, and −5.1 kcal/mol, respectively. The average length of bacterial repeats was 28.86 bp, with the longest being 36 bp and the shortest being 23 bp; the number of bacterial repeats was 30 bp. The average GC content of bacterial repeats was 55.53%, with the highest being 92% and the lowest being 12%, and the average MFE of RNA secondary structure was −6.47 kcal/mol. The average length of medium thermophile repeats was 30.24 bp, with the longest being 37 bp and the shortest being 23 bp; the number of medium thermophile repeats was 30 bp. The average GC content of repeats was 39.69%, with the highest being 79.17% and the lowest being 12%, and the mean MFE of RNA secondary structure was −4.50 kcal/mol. The average length of the repeats was 28 bp, with the longest being 36 bp and the shortest being 23 bp; the number of repeats was 25 bp. The average GC content of the repeats was 40.47%, with the highest being 68.97% and the lowest being 31.03%. The average MFE of the RNA secondary structure was −4.11 kcal/mol. RNA secondary structures and minimum free energy (MFE) of all repeat sequences predicted by the RNA fold web server are shown in Appendix A. In addition, the overview of RNA secondary structure of repeat sequences is demonstrated in Figure 3. Six super classes (A–F) were formed from the 209 completely different repeat sequences, and the results are displayed in Appendix A. There were 82 super class A, 34 super class B, 4 super class C, 38 super class D, 14 super class E, and 5 super class F, while 32 could not be classified as a super class. In total, 22 species were classified as Family 1, 66 species as Family 2, 6 species as Motif 3, 3 species as Motif 4, 1 species as Family 5, 1 species as Family 6, 6 species as Family 8, 6 species as Motif 11, and 2 species as Family 12. Additionally, 96 of them could not be classified as a family.

The conservative analysis of the repeat sequences in the CRISPR loci of several thermophiles revealed that their repeats varied substantially among the moderate thermophiles, thermophiles, and hyperthermophiles. The consensus sequence of the repeats of different thermophiles was used to predict the secondary structure of RNA, and multiple sequences were contrasted (Figure 4).

### 3.4. Spacer Sequence Analysis

The number of spacer sequences of different lengths were performed by a script written in R language. As presented in Figure 5, a total of 7744 spacer sequences, with lengths ranging from 30 to 52, were identified. Of these, the moderate thermophiles had 2290 spacer sequences, with an average of 26.3; the thermophiles had 4996 spacer sequences, with an average of 21.7; the hyperthermophiles had 458 spacer sequences, with an average of 10.1.

### 3.5. The Relationship between the CRISPR-Cas System of Thermophilic Bacteria and the Optimum Growth Temperature

The number and the proportion of different CRISPR-Cas system types in thermophilic bacteria are illustrated in Figure 6A. The largest proportion of these was type III, followed by type I, type IV, and type V. It was speculated that the CRISPR-Cas system of thermophilic bacteria preferred IB and IIIB types, and the higher the temperature, the more significant the trend. According to Figure 6B, the distribution of the average MFE of the RNA secondary structures of the repeats was the highest in hyperthermophiles, followed by thermophiles and then moderate thermophiles. The average repeat length was highest in thermophiles, followed by moderate thermophiles and hyperthermophiles. Furthermore, the relationship between repeat length and the number of repeats in different thermophilic bacteria is depicted in Figure 6C. Moderate thermophiles had the highest average GC content in the repeat sequences, followed by hyperthermophiles and then thermophiles. The average number size of spacer sequences was maximum for moderate thermophiles, followed by thermophiles and hyperthermophiles.

## 4. Discussion

Why do thermophilic bacteria survive high temperatures? What are the cellular and molecular differences between extremophiles and ordinary microbes? These have been the questions asked by scientists around the world. The heat resistance mechanism of thermophilic microorganisms mainly focuses on cell membranes, genetic material, and macromolecular material [19,20].

The cell and bacterial membranes of thermophilic microorganisms contained lipid bilayers with C20 ether bonds when they were cultivated under relatively mild conditions. The phytane chains of the two diether membranes, however, merge to produce C40 at high temperatures as a result of the fluidity of the cell membrane creating the so-called “tetraether lipid” [21]. The length, saturation, and branch chain ratio of the lipid acyl chain could all be increased at high temperatures to enhance the stability of the cell membrane [22].

In this study, 59 of 61 thermophilic bacteria were observed to contain CRISPR loci (including questionable CRISPR loci), accounting for 96.72%, which was higher than the CRISPR-Cas system discovered by Weinberger, which was about 90% of thermophilic bacteria [23]. Additionally, we identified types I, III, IV, and V CRISPR–Cas systems. In both archaeal and bacterial genomes, type I CRISPR–Cas systems were most prevalent (64% and 60% of the loci, respectively), while putative types IV and V systems were rare (<2% overall) [24]. Archaea possess significantly more type III systems than bacteria (34% versus 25% of the complete single-unit CRISPR–Cas loci); however, they lack type II systems (13% in bacteria) [9]. The type II CRISPR–Cas systems are the rarest, absent from archaea but present in <5% of bacterial genomes [25]. In this study, we did not detect types II and VI in thermophilic bacteria.

The average number of CRISPR loci was 6.9 in moderate thermophiles, 5.15 in thermophiles, and 9 in hyperthermophiles, respectively. The average length of CRISPR loci increased with the increase in temperature gradient. In this study, the complete CRISPR-Cas system, which included the CRISPR structure and the *cas* genes, was identified in 55 strains of thermophilic bacteria (90.16%). In addition, we discovered that the thermophilic bacteria with the CRISPR-Cas system preferred subtype IB and subtype IIIB, which was consistent with our previous study demonstrating that the expression of subtypes IA and IIIB of *Thermoanaerobacter tengcongensis* varied with varying temperatures [26]. CRISPR repeat sequences were typically considered to be highly different from one other aside from the similar repeat sequences in the strains of the same species or closely related species [27,28]. Repeat sequences were transcribed into crRNAs during the expression phase of CRISPR, which was crucial for adaptive immunity. The stem-loop structure formed by repeating sequences was of great significance for mediating the interaction between foreign genes targeted by spacer sequences and proteins encoded by *cas* genes [29]. Several compensatory base changes in the stem region of some CRISPR repeats have been demonstrated to generate stable secondary structures, indicating evolution and functional conservation [5]. We discovered that the secondary structure was formed by 187 of the 209 (accounting for 89.47%) unique repeat sequences at the CRISPR location of all thermophilic bacteria. However, the average value of repeat RNA secondary structure MFE deviated from our prediction (maximum for hyperthermophiles followed by thermophiles and moderate thermophiles). The average GC content of repeat sequences did not correlate with temperature, which was consistent with the conclusion that there is an association between the optimal temperature and GC content [30]. Welch’s t-test revealed that thermophilic bacteria had, on average, more CRISPR spacer sequences per genome than mesophilic bacteria (*p* < 10^−7^). However, the variance in the number of spacer sequences per genome was greater in mesophilic bacteria than in thermophilic bacteria as revealed by F-test (*p* = 5 × 10^−3^), and the largest number of spacer sequences was observed in mesophilic bacteria [17]. The results of our analysis demonstrated that the average number of interval sequences decreased with the increase in temperature gradient.

Although there have been numerous studies on the correlation between GC content and the optimum growth temperature in prokaryotes, no consensus has been reached. Studies have revealed that the GC content of thermophilic bacteria DNA (53.2%) was higher than that of normal temperature bacteria DNA (44.9%), increasing the number of hydrogen bonds in the DNA of thermophilic bacteria and the unchain temperature of DNA and ultimately contributing to the thermal stability of genetic material of thermophilic bacteria [31]. However, other researchers have discovered that the optimum growth temperature exhibited no correlation with the GC content of the genome, but the GC content of ribosomal RNA and transfer RNA correlated with the increase in temperature [32]. In this study, we observed that the mean MFE relationship of the secondary structure of repeated RNA was the highest for hyperthermophiles, followed by thermophiles and then moderate thermophiles, providing us with new implications to study the relationship between the CRISPR-Cas system of thermophilic bacteria and temperature. In a study, the 1065 prokaryotes with complete genomes and known GC content were classified into five groups based on temperature (<30 °C, 30–40 °C, 40–50 °C, 50–80 °C, and >80 °C) corresponding to hypothermic, mesophilic, moderate thermophilic, thermophilic, and hyperthermophilic, respectively. While the average GC content of the genome was similar in the other three intermediate temperature groups, the average GC content of the genome was the highest in the cryogenic group and the lowest in the hyperthermophile group (>80 °C), respectively [29]. In this study, the average GC content was the highest for moderate thermophiles, followed by hyperthermophiles and then thermophiles. The association between the GC content of repeat sequences and the optimal temperature in the CRISPR-Cas system was not reflected by the moderate thermophilic bacteria. The primary structure, amino acid preference, distribution, and hydrophobicity of proteins were all factors in the thermophilic adaptation of thermophilic bacteria, which was also affected by increasing the ionic bond interactions and assembly density, strengthening the link network between hydrogen bonds, and prompting the formation and length of disulfide bonds. These structural or functional changes helped maintain protein stability at high temperatures [15]. Additionally, trace elements (such as tungsten) in *Pyrococcus furiosus* and other archaea had a significant impact on the growth and metabolism of the strain and were crucial in the initiation of the high-temperature glycolysis pathway [33,34].

Most thermophiles revealed the characteristics of natural competence, with the Gram-negative *Thermus thermophilus* exhibiting the highest frequency of natural transformation to date [35]. Consequently, thermophilic bacteria were more likely to resist the invasion of foreign genes than other bacteria. This may be one of the factors contributing to the high discovery rate of the CRISPR-Cas system in thermophilic bacteria.

With the advancement in biotechnology, particularly the development of microbial transcriptomics and comparative transcriptomics based on RNA sequencing, the thermotolerance mechanism of thermophilic bacteria has been further explored. In addition to the immune system, other functions of the CRISPR-Cas system in prokaryotes are gradually being revealed. It was discovered that when *Thermotoga maritima* and *Caldicellulosiruptor saccharolyticus* were co-cultured, the expression of two CRISPR-related proteins increased in pure culture [36]. The researchers observed that about 90% of thermophilic bacteria had CRISPR-Cas systems whereas only 46% of mesophilic bacteria had CRISPR-Cas systems, and thermophilic archaea were also more likely to have CRISPR-Cas systems than mesophilic archaea. The presence of CRISPR-Cas was significantly linked with thermophilia in all prokaryotes. Whether the thermophilic organisms were bacteria or archaea, multiple logistic regression demonstrated a significant correlation between thermophilic and CRISPR-Cas [9]. Overall, we speculate that the CRISPR-Cas system may also be one of the mechanisms of thermophilic adaptation in thermophilic bacteria.

## 5. Conclusions

In this study, the relationship between the structural characteristics, diversity, and distribution properties of CRISPR-Cas systems and the optimum growth temperature in 61 species of thermophilic bacteria with whole-genome sequencing were statistically analyzed. The results revealed that 59 strains of 61 thermophilic bacteria had at least one CRISPR locus, accounting for 96.72% of the total. Additionally, a total of 362 CRISPR loci, 209 entirely distinct repetitive sequences, 131 *cas* genes, and 7744 spacer sequences were discovered. The average number of CRISPR loci and the average MFE of the RNA secondary structure of repeat sequences were positively correlated with temperature whereas the average length of CRISPR loci and the average number of spacers were negatively correlated. The average number of CRISPR loci, the average length of the repeats, and the GC content of the repeats did not reflect temperature dependence. The analysis and comparison of the CRISPR-Cas system of thermophilic bacteria can lay a new foundation for the study of their thermophilic adaptation mechanism.

## Figures and Tables

**Figure 1 microorganisms-11-02275-f001:**
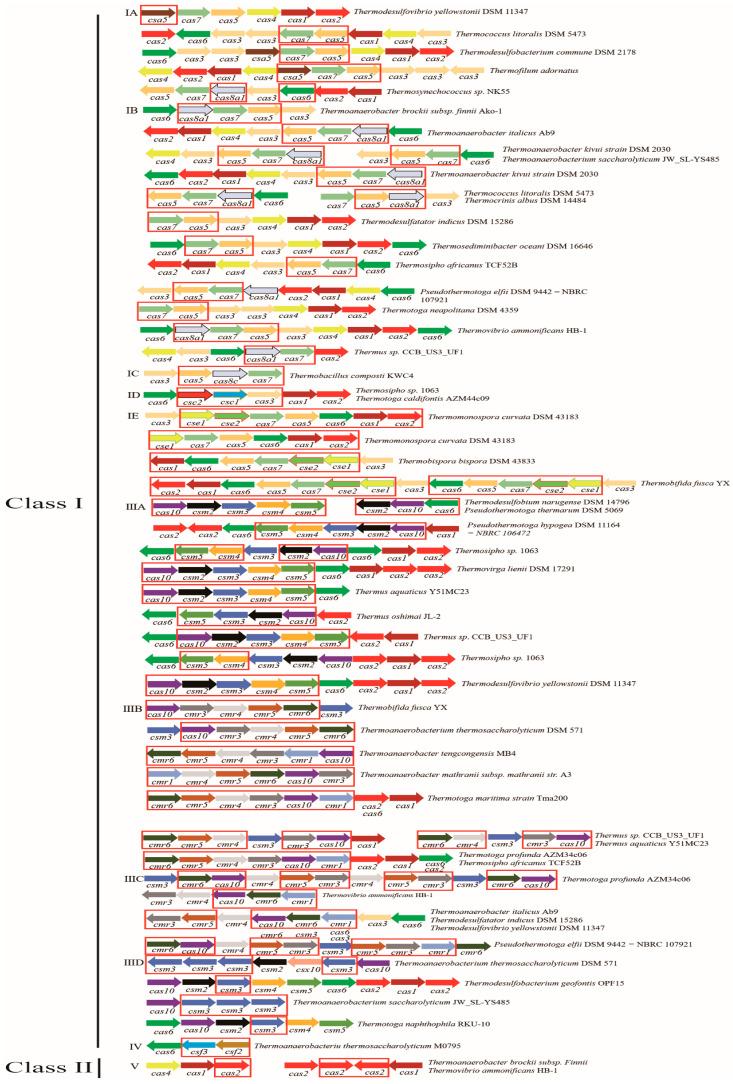
*Cas* genes clusters in 55 strains of thermophilic bacteria. The red boxes indicate marker genes of this type, and arrows of different colors and directions indicate different *cas* genes.

**Figure 2 microorganisms-11-02275-f002:**
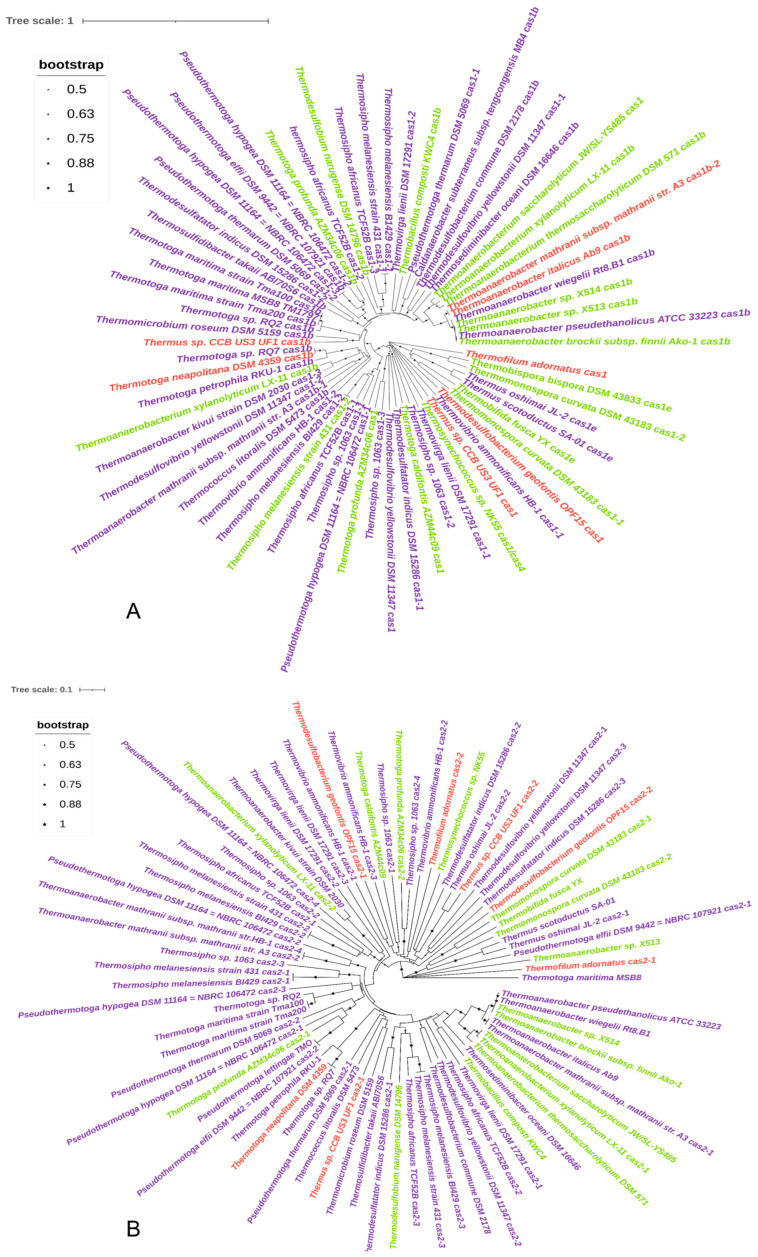
The phylogenetic tree constructed based on Cas1 and Cas2. (**A**): The phylogenetic tree constructed based on Cas1; (**B**): The phylogenetic tree constructed based on Cas2; Green represents moderate thermophiles; purple represents thermophiles; red represents hyperthermophiles.

**Figure 3 microorganisms-11-02275-f003:**
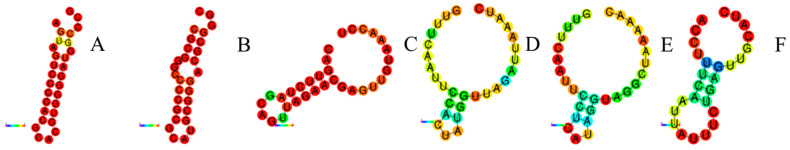
The secondary structure of repetitive RNA. (**A**,**B**) The secondary structure of repetitive RNA with minimum MFE (−15.50 kcal/mol); (**C**) The secondary structure of repetitive RNA with mean MFE (−5.1 kcal/mol); (**D**–**F**) The secondary structure of repeated RNA with the largest MFE (−0.20 kcal/mol).

**Figure 4 microorganisms-11-02275-f004:**
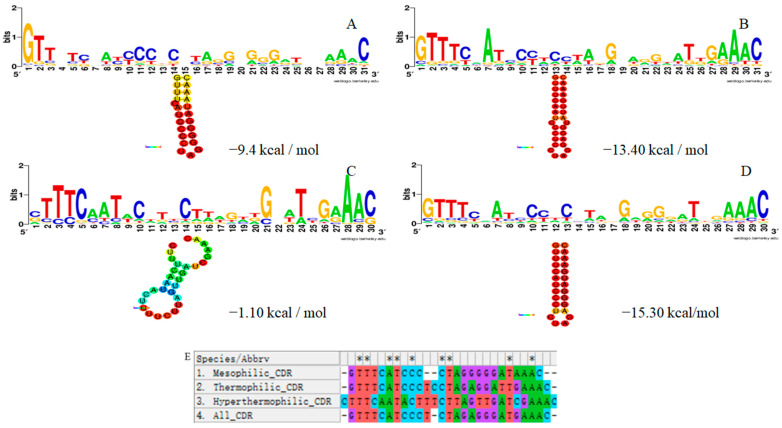
The analysis of the consensus sequence of the repeats of different thermophilic CRISPR loci. (**A**): Moderate thermophiles; (**B**): Thermophiles; (**C**): Hyperthermophiles; (**D**): All repeating sequence; (**E**): The comparison results of identical sequences of different thermophilic repeats.

**Figure 5 microorganisms-11-02275-f005:**
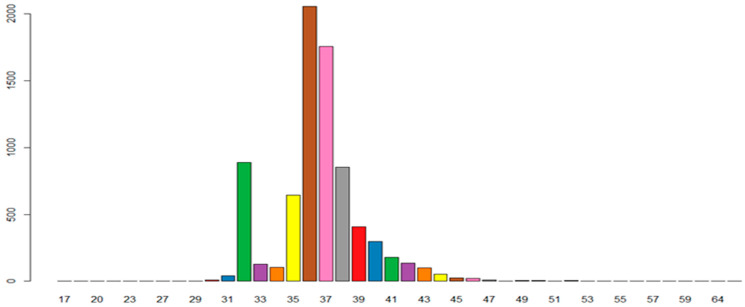
The number of spacer sequences of different lengths.

**Figure 6 microorganisms-11-02275-f006:**
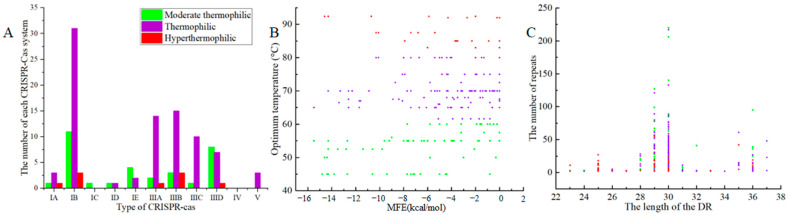
The relationship between thermophilic CRISPR-Cas system and temperature. (**A**) The type and quantitative relationship of CRISPR-Cas system in different thermophilic bacteria; (**B**) The minimum free energy of RNA secondary structure corresponding to different thermophilic bacterial repeats; (**C**) The repeat sequence of different lengths and the corresponding number of repeats. (The green represented moderate thermophiles, purple represented thermophiles, and red represented hyperthermophiles).

**Table 1 microorganisms-11-02275-t001:** CRISPR-Cas system types and proportions of moderate thermophiles, thermophiles, and hyperthermophiles.

Type	Subtype	Moderate Thermophiles	Total (%)	Thermophiles	Total	Hyperthermophiles	Total (%)
I	IA	1 (0.76%)	18 (13.74%)	3 (2.29%)	37 (28.24%)	1 (0.76%)	4 (3.05%)
	IB	11 (8.40%)		31 (23.66%)		3 (2.29%)	
	IC	1 (0.76%)		0		0	
	ID	1 (0.76%)		1 (0.76%)		0	
	IE	4 (3.05%)		2 (1.53%)		0	
III	IIIA	2 (1.53%)	14 (10.69%)	14 (10.69%)	46 (35.11%)	1 (0.76%)	5 (3.82%)
	IIIB	3 (2.29%)		15 (11.45%)		3 (2.29%)	
	IIIC	1 (0.76%)		10 (7.63%)		0	
	IIID	8 (6.11%)		7 (5.34%)		1 (0.76%)	
	IV	1 (0.76%) 3 (2.29%)		0		0	
	V			3 (2.29%)		0	
total		36		86		9	

**Table 2 microorganisms-11-02275-t002:** The length and number of repeat sequences of different thermophilic bacteria.

Category Repetitive Sequence Length	Moderate Thermophiles	Thermophiles	Hyperthermophiles	Total
23	8 (2.21%)	2 (0.55%)	2 (0.55%)	12 (3.31%)
24	6 (1.66%)	3 (0.83%)	0 (0.00%)	9 (2.49%)
25	4 (1.66%)	4 (1.10%)	17 (4.70%)	25 (6.91%)
26	3 (0.83%)	3 (0.83%)	1 (0.28%)	7 (1.93%)
27	1 (0.28%)	1 (0.28%)	1 (0.28%)	3 (0.83%)
28	6 (1.66%)	6 (1.66%)	1 (0.28%)	13 (3.59%)
29	27 (7.46%)	22 (6.08%)	8 (2.21%)	57 (15.75%)
30	44 (12.15%)	130 (35.91%)	10 (2.76%)	184 (50.83%)
31	4 (1.10%)	6 (1.66%)	0 (0.00%)	10 (2.76%)
32	3 (0.83%)	5 (1.38%)	0 (0.00%)	8 (2.21%)
33	0 (0.00%)	2 (0.55%)	1 (0.28%)	3 (0.83%)
34	0 (0.00%)	1 (0.28%)	0 (0.00%)	1 (0.28%)
35	0 (0.00%)	5 (1.38%)	1 (0.28%)	6 (1.66%)
36	5 (1.38%)	13 (3.59%)	3 (0.83%)	21 (5.80)
37	0 (0.00%)	3 (0.83%)	0 (0.00%)	3 (0.83%)
total	111 (30.66%)	206 (56.91%)	45 (12.43%)	362

## Data Availability

No new data were created or analyzed in this study. Data sharing is not applicable to this article.

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
