# Peer review of "Comparison of Structural Features of CRISPR-Cas Systems in Thermophilic Bacteria"

_microorganisms, 2023, doi:10.3390/microorganisms11092275_

Round 1

Reviewer 1 Report

In the paper entitled “Comparison of structural features of CRISPR-Cas systems in thermophilic bacteria” (control no. microorganisms-2529404) the Authors analyzed the features of clustered regularly interspaced short palindromic repeat (CRISPRs) and of CRISPR-Cas systems, and the correlation of these with the optimum growth temperature in 61 species of thermophilic bacteria. Whole-genome sequences were downloaded from GenBank and CRISPRs/CRISPR-Cas loci analyzed with CRISPRCas Finder, MEGA X, RNA fold web server, CRISPRmap, Weblogo and R script. The analysis indicated that 59 out of 61 strains contained at least one CRISPR locus, with a total of 362 CRISPR loci, 131 cas genes and 7744 spacer sequences found. Furthermore, the average number of CRISPR loci and the average minimum free energy of the RNA secondary structure of repeat sequences were positively correlated with temperature, whereas the average length of CRISPR loci and the average number of spacers were negatively correlated.

The study provides a valuable addition to the current knowledge of CRISPR-Cas system in thermophilic bacteria. Overall, the study is sound, I have only minor comments that could further improve the quality of the manuscript:

-Line 108: R script? Was an “in house” script? Please add info

-Line 115, Line 117, Line 121: “including suspected CRISPR loci” please provide an explanation, what is the criteria used to distinguish CRISPR loci by suspected CRISPR loci?

-Lines 121-125: add info on the growth temperature profile of each strain mentioned (moderate, thermophilic or hyperthermophilic strain?); Thermobifida fusca YX in table S1 contains 17 CRISPR, this is in contrast with what it is reported in the text (21), please correct accordingly

-Line 134: what is the meaning of “representative strains”?

-Line 166-167: “removing the identical duplicate sequences” please clarify the statement

-Line 216, 265: “fully functional”, these are complete CRISPR-Cas systems, functionality has not been addressed

-Line 215-224: repeated information of previous result, please rephrase to reduce redundancy

-Lines 351-354: these two sentences are repeating the same concept, please correct

-References are needed at Lines 244-245, Line 257, Line 318; furthermore, at Line 255: reference number (17) is not concordant with Weinberg et al., please correct

-Figure legend and tables lack information: Table 1: an explanation for the percentages and the total number should be added. Figure 2 and Figure 5: more explanation should be added to the legend to describe the figure and how it was generated. Supplementary Table 2: an explanation for MFE values should be added

-Several typos are present in the text, please correct: Line 18: sequencing > sequences; Line 87: bio-technology > biotechnology; Line 91: thermophilic bacteria (not in italics); Line 99, 102, 107: softwwere > software (also https://www.megasoftware.net); Line 120: strain (not in italics); Line 167: “23-37” (change color of text); Line 194: “66” write as a word; Line 195: specie > species; Line 248: the pythane (“the” capital letter); Line 122: the most CRISPR structures (highest number?); Line 126: average of 6.9 (per strain?); Line 129: 14581 and 1446 add commas and make it uniform through the text; Figure 1: cas (must be italics); Supplementary table S1: stain > strain, cas (must be italics), RefSeq > GenBank accesion number, 65°C-70°C thermophiles (change color of text), optimum temperature> optimum growth temperature.

Some spelling mistakes

Author Response

Point 1:  The graphics are generally blurry and very difficult to recognize. Especially in Fig. 1, the characters are almost indistinguishable. The evolutionary tree in Fig. 2 is originally supposed to be represented by a circle, but it is deformed into an ellipse. Moreover, the strain name cannot be determined at all.
Response 1: Thank you to the reviewer for your valuable comments. We have been modified figure 1 and figure 2. We believed that this modification makes it easy for readers to recognize images.

Point 2: In Supplementary Table 1, "NZ_CP010822.1 GCF_001399775", there is a mysterious character "æ— ". In the same line, "65°C-70°C thermophiles" is in red. Does it make sense?

Response 2: Thank you to the reviewer for your careful review. It was written errors. We have been modified.

Round 2

Reviewer 2 Report

Please confirm the attached file.

Author Response

Point 1: L123“Two moderately thermophilic strains, Thermus thermophilus JL-18 and Thermus parvatiensis strain RL,”Although described as above in the text, both strains are classified as "thermophiles" in Table S1. Are there any contradictions?

Response 1: Thanks. “moderately thermophilic strains” has been modified to "thermophiles". (Line123) 

Point 2: L230-231, The RNA fold 230 Web serve→the RNA fold 230 Web server

Response 2: Thanks, Agreed. It has been modified according to the suggestion. (Line230-231) 

Point 3: Wouldn't it be more appropriate to describe the Supplementary Figures and Tables as Figure S1 and Table S1?

Response 3: Agreed, It has been modified according to the suggestion.( Line92, 116, 231, 234)